# Improving the Structural Changes, Electrophoretic Pattern, and Quality Attributes of Spent Hen Meat Patties by Using Kiwi and Pineapple Extracts

**DOI:** 10.3390/foods11213430

**Published:** 2022-10-29

**Authors:** Heba H. S. Abdel-Naeem, Amal G. Abdelrahman, Kálmán Imre, Adriana Morar, Viorel Herman, Nabil A. Yassien

**Affiliations:** 1Department of Food Hygiene and Control, Faculty of Veterinary Medicine, Cairo University, Giza 12211, Egypt; 2Department of Animal Production and Veterinary Public Health, Faculty of Veterinary Medicine, University of Life Sciences “King Michael I of Romania”, 300645 Timișoara, Romania; 3Department of Infectious Diseases and Preventive Medicine, Faculty of Veterinary Medicine, University of Life Sciences “King Michael I of Romania”, 300645 Timişoara, Romania

**Keywords:** spent hens’ meat, tenderization, physicochemical, organoleptic, electrophoretic pattern, topographical structure

## Abstract

Spent broiler hen meat is sold at a lower price, owing to its poor texture and lower acceptability, in comparison with broiler meat. The tenderization of spent hen meat using kiwi and pineapple extracts will encourage meat processors to use this less expensive meat as a new source of raw materials for the production of different chicken meat-derived products, particularly when solving the problem associated with a great shortage of raw chicken meat materials. Therefore, the present study was undertaken to investigate the effect of kiwi extracts (5 and 7%), pineapple extracts (5 and 7%), and a combination between the two (5% kiwi and 5% pineapple) on the structural changes, electrophoretic pattern, and quality attributes of spent hen meat patties. The results demonstrated that all extract-treated meat patties exhibited a significant decrease in collagen content and shear force value, a significant increase in collagen solubility percentage, and significant improvements to all sensory attributes, in contrast to their counterpart control samples. Additionally, a non-significant change in lightness and yellowness values and a significant increase in redness value were observed in all extract-treated specimens. Moreover, the treatment of spent hen meat patties with kiwi and pineapple extracts resulted in marked degenerative changes of the muscle fiber and connective tissue, as well as a decrease in protein bands, with subsequent enhancement in tenderness. The effect was more highlighted in specimens treated with pineapple extracts (7%) and with kiwi (5%) and pineapple extracts mixture (5%).

## 1. Introduction

The tremendous growth of the poultry industry led to a wider availability of meat layers at the end of their economically productive life. Spent broiler hens produce heavy broiler birds with an adequate amount of breast and thigh meats [1]. Moreover, their meat is cheap and healthy, characterized as sources rich in protein omega-3 fatty acids, in addition to having a low cholesterol content and no religious or cultural restrictions on their consumption [2]. Additionally, they are a potentially functional type of food due to their contents that include affordable protein and bioactive peptides that possess important antioxidant, anticancer, anti-inflammatory, antihypertensive, and immunomodulatory properties [3,4,5]. However, this type of meat is tough, due to the age-related increase in heat-resistant connective tissue cross-linking, a fact that has led to the prohibition of its use, consequently, reducing its market value. Accordingly, the hens are usually slaughtered after their production period has come to an end and used in the feed and pet food production industry [6].

The disposal of unutilized spent broiler hens is considered one of the main environmental and economical burdens of the poultry industry. In this sense, the development of comminuted meat products from such meat offers an important avenue for their profitable disposal. Therefore, the production of comminuted meat products, as well as the commercialization of unconventional, less valuable spent broiler hen meat should be highlighted due to the increase in broiler meat availability [7]. Among the comminuted meat products, meat patty can achieve an excellent market demand, due to its popularity, especially in fast food restaurants.

Attempts of processing meat products from spent hen meat have been reported by Indumathi et al. [8]. However, their popularity is still very low, due to the toughness issue. In this context, the toughness of spent broiler hen meat makes it a demanding material for the production of an admissible meat product. From this perspective, it is very important to develop methods that can improve the quality of such meat, increasing its ensuing value and making it suitable for further production of different meat products. Various tenderization techniques can be used to tenderize the unacceptably tough spent hen broiler meat. Among these methods, plant-origin exogenous protease enzymes, such as papain, ginger, and bromelain, are the most reliable and popular meat tenderizers [9].

The tenderization of spent hen meat using natural sources, such as fruits, will encourage the meat processors to use this meat as a new source of materials for manufacturing of various chicken meat products, especially to solve the problem associated with a great shortage of raw chicken meat materials. The pineapple fruit has excellent juiciness and flavoring properties and can be used as a tenderizing substance in meat due to its bromelain content [10]. Likewise, it is rich in ascorbic acid, flavonoids, and phenolic compounds that possess good health-promoting antioxidants. Furthermore, it can be used as a nutritional additive because of its anti-inflammatory, anti-thrombotic, immunogenic, and tumor cell reproduction inhibitory properties [11]. Kiwi is another fruit that has a great meat tenderizing effect, due to its actinidin proteolytic enzyme content, as well as containing a variety of flavonoids and carotenoids that have demonstrated antioxidant activity [12].

Most of the previous studies focused on using kiwi and pineapple extracts for the tenderization of spent hen meat [13,14]. Nonetheless, there is limited availability of information regarding the use of these extracts for enhancing the quality features of such meat before its incorporation as raw meat material in spent hen products. In addition, owing to the lower acceptability, it correlated with the toughness of processed spent hen meat products, in comparison with broiler meat products that have been indicated by Indumathi et al. [8] and Hossain et al. [15]. Accordingly, the aim of the current investigation was to include the extracts of kiwi, pineapple, and their combination as tenderizing agents in the formulation of spent hen broiler meat patties in order to improve their physicochemical criteria, electrophoretic pattern, and sensory characteristics during frozen preservation at −18 °C for a period of 3 months.

## 2. Materials and Methods

### 2.1. Experimental Design

A three replicate-based experiment (three samples per replicate) was performed to evaluate the effect of kiwi extracts (5 and 7%), pineapple extracts (5 and 7%), and their combination (5% kiwi and 5% pineapple) on the structural modifications, electrophoretic pattern, and quality properties of the product during the formulation of spent hen meat patties.

### 2.2. Extracts Preparation

Fresh pineapple and kiwi fruits were purchased from a local supermarket in Cairo, Egypt. Each fruit was de-peeled, chopped into slices, and blended with an equal volume of chilled distilled water, for 1–2 min. The resulting specimen of each fruit was then filtered through four layers of muslin cloth and the filtrate was recovered as raw fruit extract. This crude extract was used as a potential source of proteolytic enzymes for further application in meat [16].

### 2.3. Preparation of Burger Ingredients

Ten dressed carcasses of spent hen (4 kg each) were purchased for each replicate from a local spent hen plant in Cairo, Egypt. The carcasses were transferred to the lab and stored overnight at 4 °C, for 24 h. The breast and thigh muscles were then excised from the carcasses and trimmed of obvious fat and connective tissue, then cut into small cubes (27 cm^3^) of about 100 g. Sodium tripolyphosphate and seasonings mix were acquired from Loba Chemie, Mumbai, India while the sodium chloride and starch were purchased from a local market in Cairo, Egypt.

### 2.4. Marination Process

Five groups were prepared by immersion of the spent hen meat cubes into different marinade solutions. The first and second groups were immersed in 5% and 7% concentration kiwi extract. The 3rd and 4th groups were immersed in 5% and 7% concentration pineapple extract while the 5th group was immersed in a mixture of 5% kiwi extract and 5% pineapple extract. The control group was immersed in plain water. All groups were kept at 4 °C, for 24 h. The choice of kiwi and pineapple concentration levels used in this study was inspired from preliminary experiments described in previously conducted studies, which demonstrated a good tenderizing effect of these concentrations.

### 2.5. Burger Formulation, Processing, and Storage

The marinated spent hen meat cubes (75%) from each group, as well as the control, non-marinated meat cubes, were ground with a plate grinder (4.5 mm, Seydelmann NW 114 E; Stuttgart, Germany), and then blended with water (18%), salt (1.5%), starch (5%), polyphosphates (0.3%), and seasonings (0.2%). Each mixture was shaped into burger patties (about 80 g and 1.5 cm thick) by means of a standard burger maker (9 cm inside diameter), embedded in plastic wrapping films, kept at −40 °C for 35 min, placed in plastic recipients, and then stored at −18 °C for 3 months. For each replicate, samples were taken from each group for 2nd day (zero time) and monthly examinations.

### 2.6. Product Investigations

#### 2.6.1. Quality Attributes

##### Measurement of Collagen Content and the Solubility of Collagen%

Collagen content and collagen solubility% of spent hen meat patties from each group were analyzed in agreement with the method described by Naewbanij et al. [17].

##### Measurement of Shear Force

The shear force of spent hen meat patties from each group was determined using the method recommended by Shackelford et al. [18], by means of a Warner–Bratzler shear device attached to an Instron Universal Testing Machine (Instron Corp., Model 2519-105; Canton, MA, USA).

##### Color Evaluation

The color values of spent hen meat patties from each group were measured by using a Chroma Meter (Konica Minolta, CR 410; Marunouchi, Japan), with an illuminant D_65,_ the aperture size of 11 mm, an observer of 10°, and calibrated against a white calibration plate, using the method outlined by Shin et al. [19]. From each sample, three readings were quantified and the mean values were expressed as lightness (L*), redness (a*), and yellowness (b*).

##### Sensory Panel Analysis

Sensory characteristics of the spent hen meat patties samples were analyzed following the AMSA guidelines [20]. Twenty-nine experienced panel members were selected from the Food Hygiene and Control Department from Cairo University, Egypt. Panelists were chosen based on their sensory evaluation expertise of meat products (experienced burger eaters). Furthermore, they took a preparatory session connected to a descriptive profile of sensory scores before testing; therefore, each panelist could thoroughly clarify and discuss each sensory score to be assessed. A sensory examination was performed in a special odor and noise-free room with adequate lighting and controlled temperature. Tap water was provided between each sample taken to rinse the palate. Five spent hen meat patties from each group were cooked in a forced draught oven (Heraeus D-63,450; Hanau, Germany), at 180 °C, with a central temperature of 75 °C. In a random sequence, each panelist evaluated three replicates from all groups and was requested to assign a numerical value from 1 (extremely unacceptable) to 9 (extremely acceptable) for appearance, color, flavor, tenderness, juiciness, and overall acceptability.

#### 2.6.2. Structural Examination

##### Light Microscope

Spent hen meat patties samples (1 × 1 cm) from each group were fixed for 24 h in 10% formalin and prepared and stained with Hematoxylin and Eosin, using the method described by Banchroft et al. [21]. At least 45 micrographs were examined from each group to highlight the changes that have occurred after various types of treatments.

##### Scanning Electron Microscope

Spent hen meat patties samples were sliced into small parts (2 × 2 × 3 mm) and directly fixed in 2.5% phosphate-buffered glutaraldehyde, at 4 °C, for 2 h. Each sample was washed using phosphate-buffered saline (0.1 M), three times (30 min per time), then dehydrated in ethanol (50–100%, 15 min per concentration), followed by drying for 15 min, using a critical point drier (Samdri PVT-3D; Tousimis, Rockville, MD, USA). Finally, samples were coated with gold in a vacuum evaporator (JFC 1100 E; Jeol Ltd., Tokyo, Japan) and were inspected for their topographical structure, using a scanning electron microscope (JSM 5300; Jeol Ltd., Tokyo, Japan) [22].

#### 2.6.3. Electrophoresis

Sodium dodecyl sulphate–polyacrylamide gel electrophoresis (SDS–PAGE) was performed using an Omni PAGE Maxi, Stock (VS20-48-1) (Cleaver Scientific, Rugby, United Kingdom) electrophoresis system [23]. Five grams of spent hen meat patties samples were assorted with 50 mL sodium phosphate buffer (0.01 N, pH 7.0), containing 1% 2-mercaptoethanol and 1% SDS, followed by incubation at 37 °C for a period of 2 h. The resulting mixture was centrifuged for 30 min at 1500× *g* and the supernatant was dialyzed against sodium phosphate buffer (0.1 N) containing 0.1% 2-mercaptoethanol. Dialyzed solution (50 µm) was taken for loading the gel and electrophoresis was conducted at constant voltage mode (100 V/slab and 30 mA) for 5–6 h until the dye reached the lower point of the gel. After that, the gel was removed and subsequently stained with Coomassie blue (4–5 h), destained, and then photographed.

### 2.7. Statistical Analysis

The obtained results were statistically analyzed using the SPSS statistics 17.0 for windows (IBM Corp., Armonk, NY, USA). The one-way analysis of variance (ANOVA) test was used in order to compare the data and the least square difference (LSD) test was applied to determine the significance, since the differences were considered significant at *p* < 0.05 levels.

## 3. Results and Discussions

### 3.1. The Effect of Kiwi and Pineapple Extracts on the Quality Attributes of Spent Hen Meat Patties

#### 3.1.1. The Consequence of Kiwi and Pineapple Extracts on the Collagen Content, Collagen Solubility %, and Shear Force of Spent Hen Meat Patties

The obtained results revealed that treatment of spent hen meat patties with kiwi (5 and 7%), pineapple (5 and 7%), and a mixture of kiwi (5%) and pineapple (5%) induces a significant (*p* < 0.05) reduction in the collagen content and shear force value, along with a significant (*p* < 0.05) increase in the collagen solubility %, as compared to their counterpart control samples (Table 1). In this respect, all extract-based treatments enhance the tenderness of spent hen meat patties and the effect was more noticeable in samples treated with 7% pineapple and in samples treated with the kiwi (5%) and pineapple (5%) extracts mixture.

Meat tenderness is significantly connected to the collagen content, collagen solubility, and shear force value, which can influence consumer preference quality of such meat [9,24]. Our results are in accordance with those reported by Kadıoğlu et al. [25], who observed that the treatment of spent hens breast, drumstick, and thigh meats with pineapple fruit juice at a marination time of 160 min resulted in a significant decrease of the shear force value. In another investigation, Hussain et al. [26] found that the treatment of broiler breast with pineapple core extract, using 100% maceration concentration, for 35 min, induced an 86% reduction in toughness. In this regard, pineapple extract has great potential as a meat tenderizer. Furthermore, Bagheri Kakash et al. [27] reported that kiwi had a significant consequence on the shear force and the reduced hardness of Ross breed chicken meat, owing to the activity of photolytic enzymes on myofibrillar proteins, associated with the breaking of collagenous tissues. Accordingly, the significant reduction in the collagen content and shear force value besides the significant increase in the collagen solubility % in all extract-treated samples, with subsequent improvement of their tenderness reported in the present study, can probably be related to the presence of bromelain and actinidin proteolytic enzymes in pineapple and kiwi fruits, respectively.

In this sense, spent laying hen meat tenderized with pineapple and kiwi extracts is a great alternative to other raw meats used to develop or promote meat products, and a meat management system must be taken into consideration in order to develop these food products and increase their value. However, the toughness values of spent hen meat products were significantly affected by the type of meat. For instance, Ilayabharathi et al. [28] and Indumathi et al. [8] recorded an increase in the toughness with higher shear force values in spent hen sausages, when compared to broilers. Furthermore, a significant increase in the soluble collagen content, solubility, and proteolysis rate, as well as a gradual decrease in shear force values of spent hen meat during postmortem aging, with a significant enhancement in the tenderness of spent hen minced patties prepared from such meat was observed by Vaithiyanathan et al. [29]. Nonetheless, Barido and Lee [30] found that spray marination using a crude protease (4%) during the postmortem tenderization mechanism of spent hen breast induced significant improvement in its protein solubility, as well as myofibrillar fragmentation index, with a non-significant effect on sarcoplasmic protein and collagen contents, suggesting that substrate hydrolysis was restricted to myofibrillar protein.

#### 3.1.2. The Effect of Kiwi and Pineapple Extracts on the Instrumental Color Values of Spent Hen Meat Patties

It is conspicuous that meat color strongly influences consumers’ choice to purchase, since it is a reference of freshness and quality [31]. The color values of spent hen meat patties treated with kiwi (5 and 7%), pineapple (5 and 7%), and a mixture of kiwi (5%) and pineapple (5%) are presented in Table 2. We also noticed a non-significant (*p* > 0.05) change in lightness (L*), as well as yellowness (b*) values, with a significant (*p* < 0.05) increase in redness (a*) value in all extract-treated samples.

The obtained results are in harmony with those obtained by Kim et al. [32], who reported non-significant changes in L* and b* value, with a significant increase in the a* value of pork jerky products marinated with kiwi (5%) or pineapple (5%) juice, when compared with untreated control samples. Moreover, Toohey et al. [33] and Jiao et al. [34] observed that the treatment of beef with kiwi fruit solution induces a non-significant change in L* and b* values. Likewise, Singh et al. [35] reported non-significant modifications in the color values of spent hen chicken emulsion incorporated with kiwi peel extract (2.5%), except the a* and b*, which only significantly increased at the end of the chilling storage (9 days), when compared with the control sample. In contrast, the marination of Ross breed chicken with kiwi extract significantly increased L* [27]. In addition, the marination of spent hen meats with pineapple fruit juice, for 80 min, significantly increased L*, a*, and b* values [25].

#### 3.1.3. The Effect of Kiwi and Pineapple Extracts on the Sensory Attributes of Spent Hen Meat Patties

The sensory scores of spent hen meat patties treated with kiwi (5 and 7%), pineapple (5 and 7%), and a mixture of kiwi (5%) and pineapple (5%) are shown in Figure 1, Figure 2 and Figure 3.

Results of sensory examination revealed that there is a significant improvement in the case of all sensory attributes (appearance, color, flavor, tenderness, juiciness, and overall-acceptability) of all extract-treated samples, as compared to their counterpart control samples. Among all extract-treated specimens, treatment of spent hen patties with a mixture of kiwi (5%) and pineapple (5%) achieved the higher sensory scores, however, lower consumer acceptability was recorded in samples treated with pineapple (7%), owing to over tenderization and the mushy texture of the product.

Our results are in agreement with those reported by Kang et al. [13] and Kadıoğlu et al. [25], who found that the marination of spent hen meat with pineapple fruit significantly improves the tenderness score. Additionally, the injection of spent hen meat with kiwi fruit enzyme (10%) improved its juiciness and tenderness, as well as overall acceptability [36]. In another study, Pooona et al. [37] obtained an enhancement in flavor, texture juiciness, and overall acceptability of spent hen chicken nuggets marinated with kiwi fruit juice. In the same regard, improvement of the sensory attributes of spent hen chicken emulsion incorporated with kiwi peel extract (2.5%) during storage at 4 °C for 9 days was reported by Singh et al. [35]. Furthermore, Jiao et al. [34] noticed that the treatment of beef with kiwi fruit improved its sensory attribute and extended its shelf life during refrigeration at 4 °C for 7 days, in contrast to the control sample, where rancid odor and a significant decrease in taste, as well as texture scores, were recorded at the end of storage. Similarly, Żochowska-Kujawska et al. [38] indicated that the addition of pineapple and kiwi fruits, during the formulation of pork dry sausage, improved its tenderness, juiciness, chewiness, and palatability.

### 3.2. The Effect of Kiwi and Pineapple Extracts on the Structural Changes of Spent Hen Meat Patties

#### 3.2.1. Light Microscope

Light micrographs images of the control-untreated spent hen meat patties, stained with H&E showed straight, nearly intact, closely bound, nucleated muscle fibers (Figure 4I(A–C)), with a massive amount of undamaged fibrous connective tissue (Figure 4I(D–F)). However, the treatment of spent hen meat patties using fresh kiwi extract (5%) consisted of the appearance of multiple longitudinal and cross cracks across the muscle fibers (Figure 4II(A–C)), with slight degenerative changes in the connective tissue (Figure 4II(D–F)).

Furthermore, light micrographs of spent hen meat patties incorporated with fresh kiwi extract (7%) displayed more degenerative changes in the muscle fibers (Figure 5I(A–C)), with a noticeable change in the fibrous connective tissue (Figure 5I(D–F)). However, the micrographs of spent hen meat patties incorporated with pineapple extract (5 and 7%), as well as spent hen meat patties incorporated with a mixture of kiwi (5%) and pineapple (5%) extract exhibited the appearance of gaps, as a consequence of severe muscle fragmentation (Figure 5II and Figure 6I,II(A–C)), in addition to the extensive degradation in the fibrous connective tissue (Figure 5II and Figure 6I,II(D–F)).

It is obvious that the investigations that suggested the effect of kiwi and pineapple extract on the histological structure of spent hen meat patties using a light microscope are very limited. Our observations are in good line with those published by Abdelrahman et al. [16], who observed many longitudinal and cross breaks in the muscle fibers, degradation of the cell membrane, and appearance of holes within muscle fibers in spent hen muscles treated with marinades containing kiwi (5 and 10%) and pineapple (5 and 10%) extracts. In addition, our results are in harmony with those reported in camel meat [9] and camel meat burgers [39] treated with ginger and papain extracts. Furthermore, Jiao et al. [34] found that kiwi fruit induces a beneficial effect on the microstructure of beef by protecting it from hydratability during chilling storage (4 °C), in contrast to the untreated control sample, which had lost its structural integrity, especially on the 3rd day of storage.

#### 3.2.2. Scanning Electron Microscope

The topographical structure description of muscle tissue using scanning electron microscope (SEM) is important to obtain information about the modifications in the external morphology, texture, or orientation of muscle fibers [31]. Scanning electron micrographs of untreated control spent hen meat patties displayed closely packed muscle fibers and narrow intermyofiber spaces in between (Figure 7A–C), with intact connective tissue fibers (Figure 7D). The topographical structure of spent hen meat patties treated with kiwi (5%) revealed coagulated muscle fibers (Figure 7E), the presence of cracks, and fragmentations (Figure 7F,G), with cord-like connective tissue (Figure 7H). However, the treatment of spent hen meat patties treated with the kiwi extract (7%), pineapple extract (5 and 7%), and a mixture of kiwi (5%) and pineapple (5%) extract resulted in several cleavages and fragmentation in the muscle fibers (Figure 8 and Figure 9A–C,E–G), with coagulated (Figure 8D,H), granulated (Figure 9D), and amorphous mass (Figure 9H) connective tissue.

These observations were in harmony with those reported by Ketnawa and Rawdkuen [22], who demonstrated that the topographical structure of chicken muscles treated with bromelain showed broken muscle fibers, loss of muscle fiber interactions, and gaps between muscle fibers. Furthermore, severe endomysium degradation in spent hen breast samples, marinated using ginger extract, with a succeeding improvement in their tenderness, was recorded by Kumar et al. [40].

Likewise, the topographical composition of chicken meat treated with lemon juice revealed irregular, broken, and separated muscle fibers [41]. In addition, the swelling of muscle fibers with an increase in muscle fiber diameter was reported in sodium bicarbonate and lactic acid marinated spent hen meat [42].

Moreover, Rumondor et al. [43] reported that treatment of spent hen meat sausage with angkak fermented rice resulted in the appearance of cavities, with loss of surface texture. However, a well-organized structure was observed by Abdel-Naeem et al. [31] together with closely packed muscle fibers with narrow intermyofiber spaces in between, alongside a dense fibrous network of connective tissue elements in untreated control chicken meat.

### 3.3. The Effect of Kiwi and Pineapple Extracts on the Electrophoretic Pattern of Spent Hen Meat Patties

The electrophoretic pattern of spent hen meat patties treated with kiwi (5 and 7%), pineapple (5 and 7%), and a mixture of kiwi (5%) and pineapple (5%) showed a reduction in the number and the intensity of protein bands that revealed an increase in the muscle proteolysis, when compared with control-untreated spent hen meat patties (Figure 10). Such an observation was more pronounced in spent hen meat patties treated with pineapple (7%) (Figure 10 (P7%)), and a mixture of kiwi (5%) and pineapple (5%) (Figure 10 (M)). These findings confirm the result regarding protein solubility in extract-treated spent hen meat patties (Table 1).

Similar to the obtained results, Abdelrahman et al. [16] reported that the electrophoretic pattern of spent hen breast muscles treated with kiwi and pineapple revealed a reduction in the number of protein bands and a reduction in band intensity, in comparison with control. The breakdown of protein bands of beef [44] and chicken [22] after treatment with bromelain from pineapple peel have been reported. In addition, the degradation of myofibrillar proteins and the appearance of new peptides have been observed in lamb treated with kiwi fruit juice [45].

In another study, Liu et al. [46] reported substantial degradation of myosin in porcine muscles treated with kiwi fruit juice. In this regard, Żochowska-Kujawska et al. [38] recorded that natural extracts of kiwi and pineapple could digest muscle proteins when they were mixed with meat, as indicated by a higher proteolysis index. Moreover, lower numbers of protein bands were observed in spent hen meat mixed with 2.5% ginger powder [47] and with ammonium hydroxide [48].

## 4. Conclusions

From the current investigation, it was observed that the treatment of spent hen meat patties with kiwi and pineapple extracts induced a significant reduction in the collagen content and shear force value, a significant increase in collagen solubility % with significant improvements to all sensory attributes, when compared with the control samples. The histological and topographical structure of all extract-treated spent hen meat patties displayed cracks and fragmentations in the myofibers, with clear degenerative changes in connective tissue. Additionally, the electrophoretic pattern of all extract-treated spent hen meat patties showed a decrease in the number and the intensity of protein bands, with an increase in muscle proteolysis. Accordingly, kiwi and pineapple extracts can be utilized by processors to increase the consumer acceptance of spent hen meat, in addition to the increase of its suitability, as a raw material, for the production of various chicken meat products.

## Figures and Tables

**Figure 1 foods-11-03430-f001:**
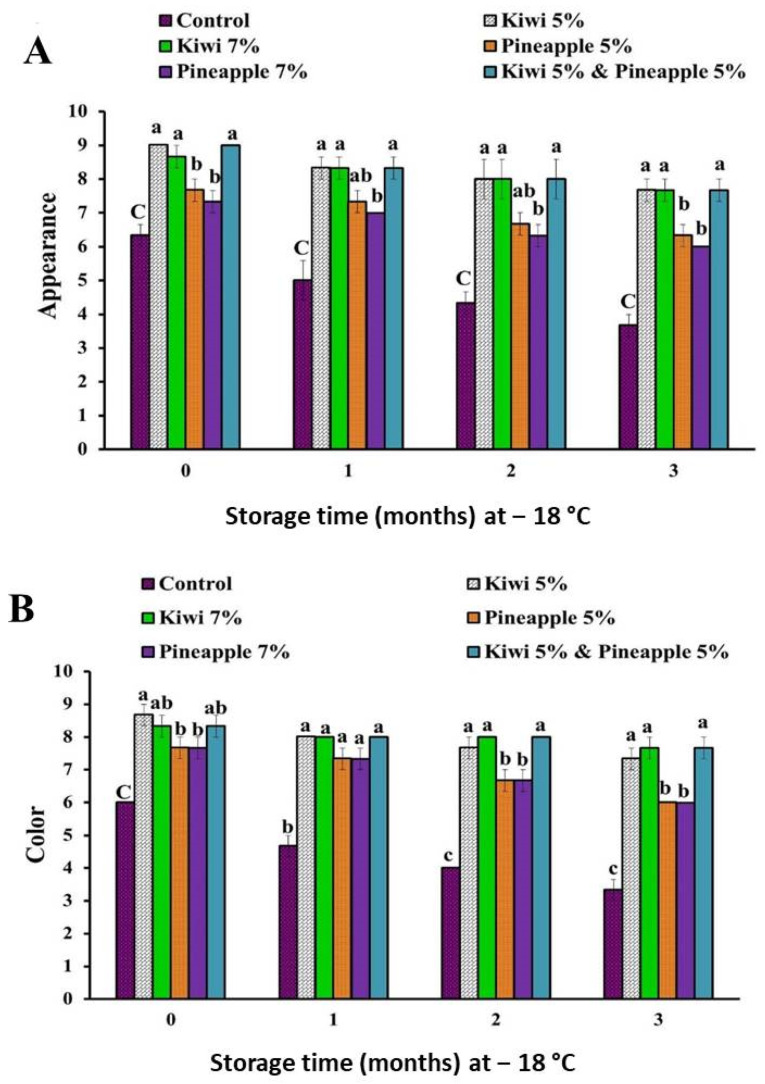
Appearance (**A**) and color (**B**) scores of spent hen meat patties treated with kiwi (5 and 7%), pineapple (5 and 7%), and a mixture of kiwi (5%) and pineapple (5%), during frozen preservation at −18 °C, for a period of 3 months. Columns with different letters within the same month of storage indicate significant differences at *p* < 0.05.

**Figure 2 foods-11-03430-f002:**
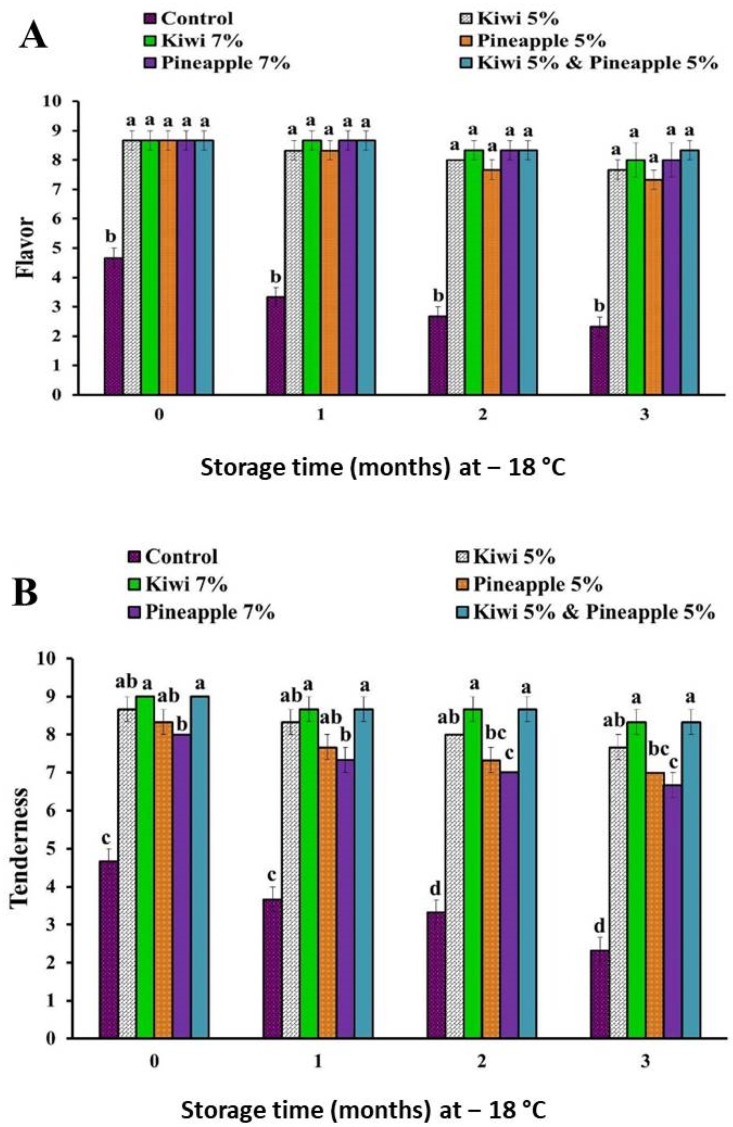
Flavor (**A**) and tenderness (**B**) scores of spent hen meat patties treated with kiwi (5 and 7%), pineapple (5 and 7%), and a mixture of kiwi (5%) and pineapple (5%), during preservation at −18 °C, for a period of 3 months. Columns with different letters, within the same month of storage indicate significant differences at *p* < 0.05.

**Figure 3 foods-11-03430-f003:**
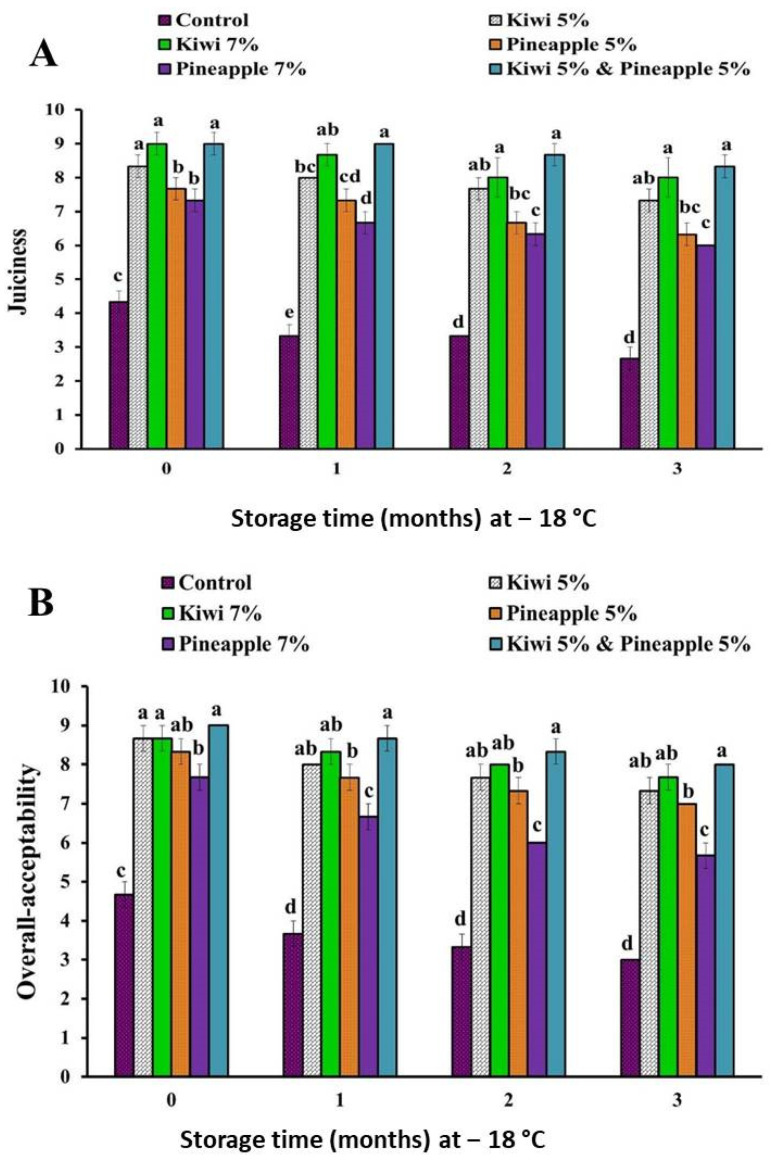
Juiciness (**A**) and overall-acceptability (**B**) scores of spent hen meat patties treated with kiwi (5 and 7%), pineapple (5 and 7%), and a mixture of kiwi (5%) and pineapple (5%) during preservation at −18 °C for a period of 3 months. Columns with different letters, within the same month of storage indicate significant differences at *p* < 0.05.

**Figure 4 foods-11-03430-f004:**
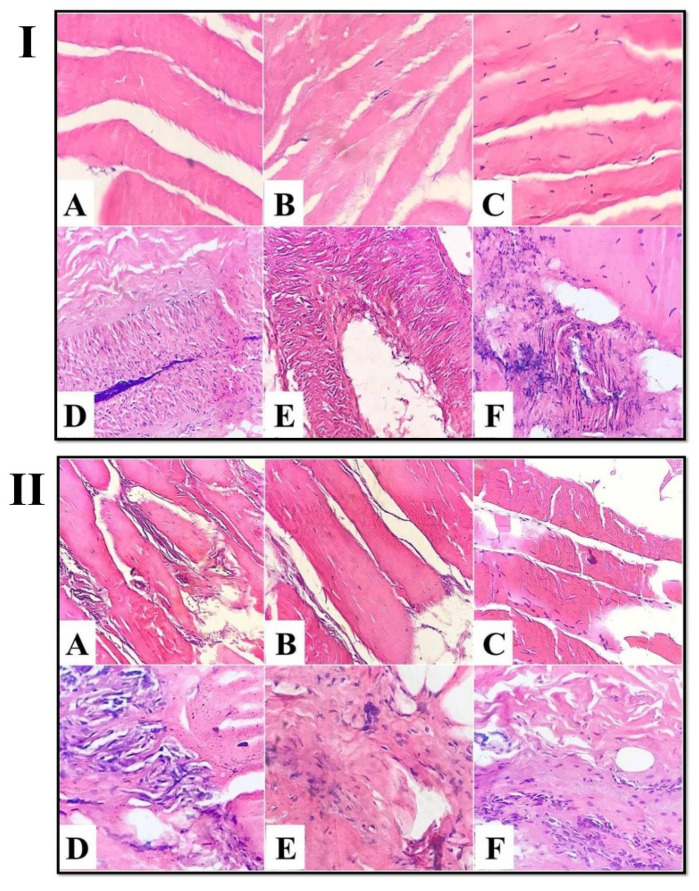
Light micrograph of control-untreated spent hen meat patties (**I**) and spent hen meat patties treated with 5% kiwi extract (**II**) stained with H&E (×40). Muscle fiber (**A**–**C**) and connective tissue (**D**–**F**).

**Figure 5 foods-11-03430-f005:**
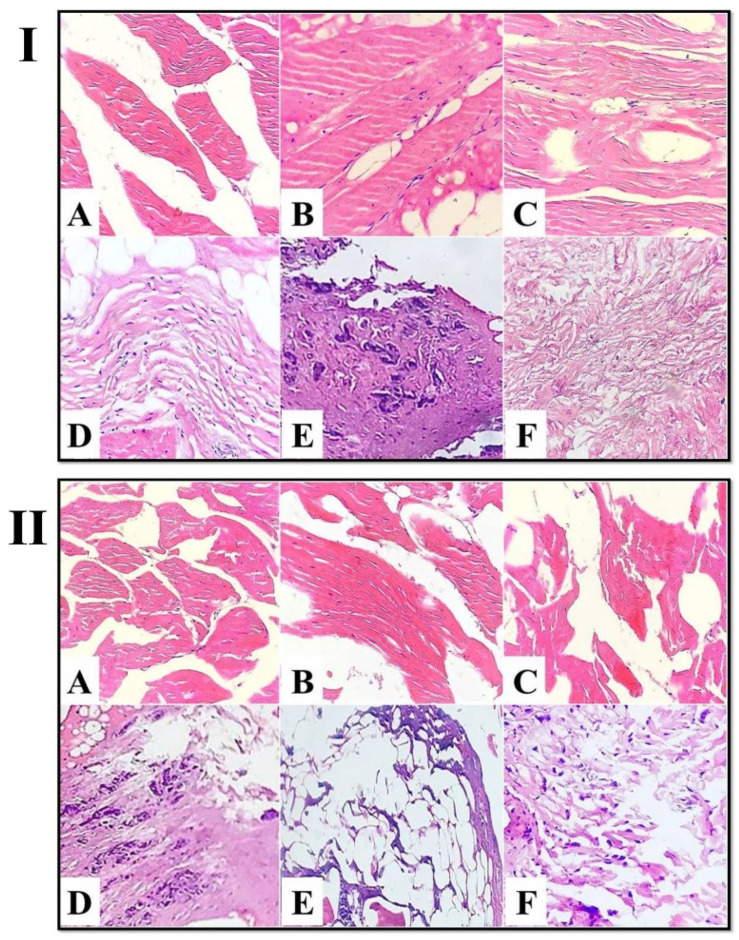
Light micrograph of spent hen meat patties treated with 7% kiwi extract (**I**) and with 5% pineapple extract (**II**) stained with H&E (ob. ×40). Muscle fiber (**A**–**C**) and connective tissue (**D**–**F**).

**Figure 6 foods-11-03430-f006:**
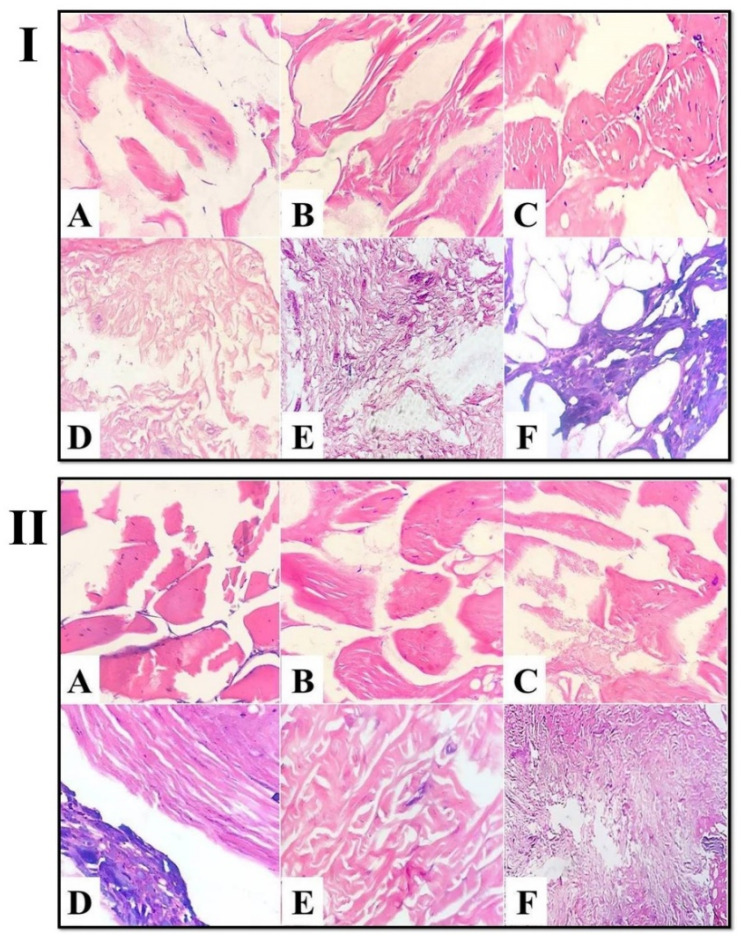
Light micrograph of spent hen meat patties treated with 7% pineapple extract (**I**) and with a mixture of kiwi (5%) and pineapple (5%) (**II**) stained with H&E (×40). Muscle fiber (**A**–**C**) and connective tissue (**D**–**F**).

**Figure 7 foods-11-03430-f007:**
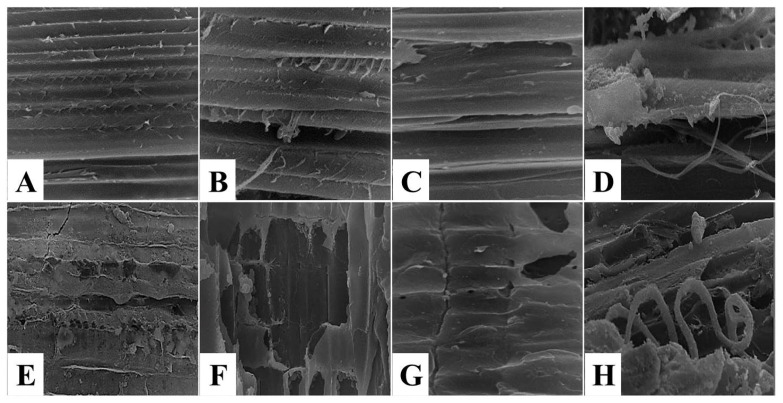
SEM micrographs of control-untreated spent hen meat patties (**A**–**D**) and spent hen meat patties treated with 5% kiwi extract (**E**–**H**). Muscle fiber with magnification ×350 (**A**,**E**) and ×1000 (**B**,**C**,**F**,**G**); Connective tissue with magnification ×1000 (**D**,**H**).

**Figure 8 foods-11-03430-f008:**
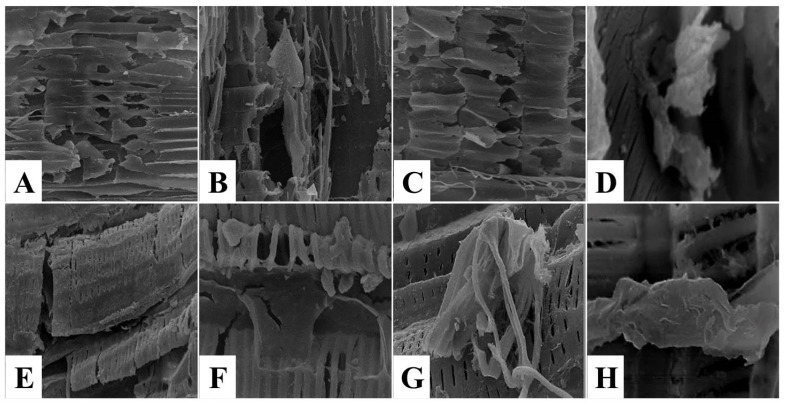
SEM micrographs of spent hen meat patties treated with 7% kiwi extract (**A**–**D**) and with 5% pineapple extract (**E**–**H**). Muscle fiber with magn. ×350 (**A**,**E**) and ×1000 (**B**,**C**,**F**,**G**); Connective tissue with magn. ×1000 (**D**,**H**).

**Figure 9 foods-11-03430-f009:**
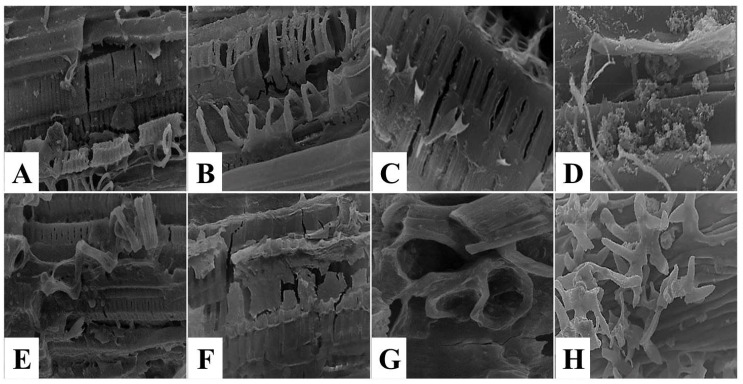
SEM micrographs of spent hen meat patties treated with 7% pineapple extract (**A**–**D**) and with a mixture of kiwi (5%) and pineapple (5%) (**E**–**H**). Muscle fiber with magn. ×350 (**A**,**E**) and ×1000 (**B**,**C**,**F**,**G**); Connective tissue with magn. ×1000 (**D**,**H**).

**Figure 10 foods-11-03430-f010:**
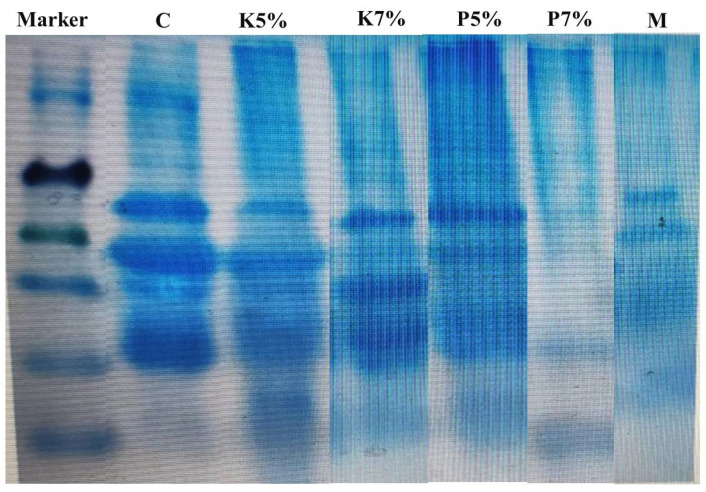
Electrophoretic pattern of control untreated-spent hen patties (C), spent hen meat patties treated with 5% kiwi (K5%), 7% kiwi (K7%), 5% pineapple (P5%), 7% pineapple (P7%), and a mixture of kiwi (5%) and pineapple (5%) (M).

**Table 1 foods-11-03430-t001:** Collagen content, collagen solubility %, and recorded values of shear force of spent hen patties treated with kiwi (5 and 7%), pineapple (5 and 7%), and a mixture of kiwi (5%) and pineapple (5%).

	Collagen Content (g%)	Collagen Solubility (%)	Shear Force (N)
Control	1.08 ^a^ ± 0.02	6.62 ^f^ ± 0.72	14.42 ^a^ ± 0.00
Kiwi 5%	0.51 ^b^ ± 0.02	10.44 ^e^ ± 0.54	11.47 ^b^ ± 0.03
Kiwi 7%	0.34 ^c^ ± 0.02	15.64 ^d^ ± 0.59	8.81 ^c^ ± 0.01
Pineapple 5%	0.25 ^d^ ± 0.00	21.40 ^c^ ± 0.20	7.94 ^d^ ± 0.01
Pineapple 7%	0.05 ^f^ ± 0.01	59.11 ^a^ ± 2.72	3.63 ^f^ ± 0.00
Kiwi 5% and Pineapple 5%	0.12 ^e^ ± 0.00	39.57 ^b^ ± 3.76	6.57 ^e^ ± 0.00

^a–f^ Means with different superscripts in the same column are significantly (*p <* 0.05) different. Values indicate the mean ± SE.

**Table 2 foods-11-03430-t002:** Instrumental color evaluation of spent hen patties treated with kiwi (5 and 7%), pineapple (5 and 7%), and a mixture of kiwi (5%) and pineapple (5%).

	L*	a*	b*
Control	58.28 ^a^ ± 0.97	6.60 ^c^ ± 0.02	11.99 ^a^ ± 0.10
Kiwi 5%	57.31 ^a^ ± 0.88	7.97 ^b^ ± 0.02	12.37 ^a^ ± 0.24
Kiwi 7%	56.43 ^a^ ± 0.92	8.67 ^a^ ± 0.03	12.38 ^a^ ± 0.21
Pineapple 5%	57.64 ^a^ ± 0.59	7.79 ^b^ ± 0.03	12.13 ^a^ ± 0.14
Pineapple 7%	57.44 ^a^ ± 0.70	7.86 ^b^ ± 0.09	12.14 ^a^ ± 0.13
Kiwi 5% and Pineapple 5%	57.46 ^a^ ± 0.77	7.89 ^b^ ± 0.01	12.00 ^a^ ± 0.35

^a–c^ Means with different superscripts in the same column are significantly (*p <* 0.05) different. Values indicate the mean ± SE.

## Data Availability

Data are contained within the article.

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
