# Peer review of "Improving the Structural Changes, Electrophoretic Pattern, and Quality Attributes of Spent Hen Meat Patties by Using Kiwi and Pineapple Extracts"

_foods, 2022, doi:10.3390/foods11213430_

Round 1

Reviewer 1 Report

Although the paper is interesting and the experimental outlay, etc is acceptable.

The paper needs major English language editing.

There is a major flaw, and in my mind,  this flaw needs to be addressed.

What is lacking is a quantification of the specific enzymes that are involved. Please provide the concentrations of the enzymes present in the different treatments (fruit extracts).

Author Response

Reviewer #1:

Although the paper is interesting and the experimental outlay, etc. is acceptable.

Thank you so much for your positive appreciation! We are delighted to read this line! Thank you so much!

Query 1: The paper needs major English language editing.

Answer 1: According to the rviewer request, the manuscript was undergo to extensively English editing. All of the resulted modifications are highlighted with red font. Please see the revised version.

Query 2: There is a major flaw, and in my mind, this flaw needs to be addressed. What is lacking is a quantification of the specific enzymes that are involved. Please provide the concentrations of the enzymes present in the different treatments (fruit extracts).

Answer 2: Thank you very much for the comment. With respect to the reviewer opinion, the determinations of enzyme concentrations in the fruit did not constitute the aim of the present work. Therefore, the research team is not able to provide such values.

Special thanks for your understanding!

Reviewer 2 Report

Dear authors,

You have presented all-rounded paper, which is with clear and easy to follow structure, sufficient and up-to-date discussion and logical conclusions. The only drawback is the sensory analysis which is not sufficiently explained about number of assessors, and their education and training, demographic structure. Furthermore, likeability testing o will also small number of assessors is not reliable source of information; consumers are preferred subject o that topic. If there are such data, then it would benefit much more. 

Author Response

Reviewer #2:

Dear authors,

You have presented all-rounded paper, which is with clear and easy to follow structure, sufficient and up-to-date discussion and logical conclusions.

Dear reviewer, Thank you so much for your review, kind comments, and valuable suggestions. We have modified the text according to them. Thank you very much for your positive comments.

Query 1: The only drawback is the sensory analysis which is not sufficiently explained about number of assessors, and their education and training, demographic structure. Furthermore, likeability testing o will also small number of assessors is not reliable source of information; consumers are preferred subject o that topic. If there are such data, then it would benefit much more.

Answer 1: All the information required is written in detail in materials and methods in the revised version of the manuscript (line 155 and lines 167-172).

Reviewer 3 Report

L 119-121 - please explain

the manuscript is very interesting for the scientists, meat industry and other producers of spent broiler hen meat. It has a great potential with gained results, i.e. kiwi and pineapple extracts can be utilized by meat processors to increase the consumer acceptance of spent hen meat and to  increase its suitability as a raw material for the production of various chicken meat  products. The experiment is conducted correctly, but I have some minor remarks (mentioned in the section for authors). After the authors revise the manuscript, i.e. give the proper explanation, the paper could be recommended for further processing. 

Why did you choose to use small cubes for marination process? Dimensions of all cubes were (cca?)..

L 129-131 - you reported that the samples were withdrawn from each group for examination on the 2nd day and monthly. Why are these results regarding time reported only for sensory attributes?

Author Response

Reviewer #3:

Query 1: L 119-121 - please explain

Answer 1: In the preliminary experiment, we use different levels of kiwi and pineapple. The higher concentrations than that used in the current study lead to over-tenderization and unacceptable product to the panelist however using lower concentrations was less effective to overcome the toughness issue related to spent hen meat.

The manuscript is very interesting for the scientists, meat industry and other producers of spent broiler hen meat. It has a great potential with gained results, i.e. kiwi and pineapple extracts can be utilized by meat processors to increase the consumer acceptance of spent hen meat and to increase its suitability as a raw material for the production of various chicken meat  products. The experiment is conducted correctly, but I have some minor remarks (mentioned in the section for authors). After the authors revise the manuscript, i.e. give the proper explanation, the paper could be recommended for further processing. Thank you for your appreciation! Great thanks for your revision and respected suggestions!

Query 2: Why did you choose to use small cubes for marination process? Dimensions of all cubes were (cca?).

Answer 2: We choose to use small cubes of meat to facilitate the penetration of fruit extract deep into meat cubes. Moreover, the dimensions of all cubes were added (27 cm3) in the revised manuscript line 119.

Query 3: L 129-131 - you reported that the samples were withdrawn from each group for examination on the 2nd day and monthly. Why are these results regarding time reported only for sensory attributes?

Answer 3: Thank you for your comments. Meat processors are concerned about the quality characteristics of processed meat products under storage. In this context, prepared spent hen meat patties were stored at −18 °C for 3 months and the quality characteristics were assessed post-processing. The most important quality attributes that could be affected during freezing storage are sensory attributes and deterioration criteria (their data under publication) due to the action of spoilage bacteria and proteolytic enzymes.